# Nontrivial Topological Properties and Synthesis of Sn_2_CoS with L2_1_ Structure

**DOI:** 10.3390/nano13081389

**Published:** 2023-04-17

**Authors:** Guifeng Chen, Bolin Long, Lei Jin, Hui Zhang, Zishuang Cheng, Xiaoming Zhang, Guodong Liu

**Affiliations:** 1Hebei Engineering Laboratory of Photoelectronic Functional Crystals, Hebei University of Technology, Tianjin 300130, China; longbolin1226@163.com (B.L.);; 2School of Materials Science and Engineering, Hebei University of Technology, Tianjin 300130, China

**Keywords:** topological nodal line, dirac point, electrochemical deposition (ECD), nanowires

## Abstract

We synthesize Sn_2_CoS in experiment and study its topological properties in theory. By first-principles calculations, we study the band structure and surface state of Sn_2_CoS with L2_1_ structure. It is found that the material has type-II nodal line in the Brillouin zone and clear drumhead-like surface state when the spin–orbit coupling is not considered. In the case of spin–orbit coupling, the nodal line will open gap, leaving the Dirac points. To check the stability of the material in nature, we synthesize Sn_2_CoS nanowires with L2_1_ structure in an anodic aluminum oxide (AAO) template directly by the electrochemical deposition (ECD) method with direct current (DC). Additionally, the diameter of the typical Sn_2_CoS nanowires is about 70 nm, with a length of about 70 μm. The Sn_2_CoS nanowires are single crystals with an axis direction of [100], and the lattice constant determined by XRD and TEM is 6.0 Å. Overall, our work provides realistic material to study the nodal line and Dirac fermions.

## 1. Introduction

In recent years, topological semimetals have received extensive research interest [1,2,3]. According to the dimension of the band crossing, topological semimetals can be divided into zero-dimensional nodal-point semimetal [4,5,6,7,8,9], one-dimensional nodal-line semimetal [10,11,12,13,14,15,16,17,18,19], and two-dimensional nodal-surface semimetal [20,21,22,23]. According to the energy band dispersion, the nodal point can actually be divided into type I and type II. The type-I nodal point has a positive cone dispersion, which completely separates electrons and holes. The type-II nodal point has a sloping cone dispersion, which can realize the coexistence of electrons and holes. Nodal lines are similar to the classification of nodal points and can also be further divided into type I, type II [24,25,26], and hybrid type [27,28]. If all band crossings on a nodal line belong to type I (type II), then the nodal line is type I (type II). If it contains both type-I and type-II band crossings, the nodal line belongs to the hybrid type. Type-II nodal lines are currently found in two-dimensional and three-dimensional materials, including K_4_P_3_ [24], Mg_3_Bi_2_ [24], pure Ti metal [26], and YCd [9]. The type-II nodal line is different from the traditional type-I nodal line. It has many interesting properties, including high anisotropic magnetoresistance, a direction-reliant chiral anomaly, and novel Klein tunneling in the momentum space [29,30,31,32,33]. In addition, nodal points can also be classified by degeneracy, including Dirac [34,35,36,37,38], Weyl [39], and triple degeneracy points [40,41]. The Dirac point is a fourfold degenerate point, which can be seen as a combination of a pair of Weyl points with opposite chirality. According to the energy band dispersion, Dirac points can also be divided into two types: type I and type II.

Condensed matter physics is different from high-energy physics and does not require Lorentz invariance, so type-II Dirac fermions are also Lorentz-violating fermions. Although the research on Dirac semimetals is very extensive, there are few studies on the type-II Dirac fermion. One major reason is that the excellent candidate materials for type-II Dirac semimetals are still limited. Another major challenge is that among the candidate materials, there are fewer materials that can be synthesized experimentally and stably exist in an ambient environment. Actually, type-II Dirac semimetals were found in XMg_2_Ag (X = Pr, Nd, Sm), XInPd_2_ (X = Ti, Zr, Hf), VAl_3_, PtSe_2_, PtTe_2_, NiTe_2,_ YPd_2_Sn, and RbMgBi [42,43,44,45,46]. Among the reported materials, only the materials including VAl_3_, PtSe_2_, PtTe_2,_ and NiTe_2_ have been confirmed in experiments. Even though the materials can be synthesized, not every material has a good measurable topological signal. The type-II Dirac points in PtSe_2_ and PtTe_2_ locate quite far away from the Fermi level (at −1 eV to −2.54 eV), which leads to the fact that the topological signal of a type-II Dirac point cannot be detected in practical application. NiTe_2_ is prepared by a two-step solvothermal technique [47], and the NiTe_2_ synthesized by this method is a polycrystalline powder with a small grain size. In this polycrystalline powder, the topological signal generated by the surface state cannot play a role in practical devices. As for VAl_3_, so far, only bulk single crystals are grown by the flux method above 1000 °C, and the synthesis time exceeds 24 h [43]. Nanowire and thin film form materials with type-II Dirac points suitable for practical applications still do not appear. Therefore, the search for new type-II Dirac metals, especially those that can be synthesized experimentally and exist in an ambient environment, is the primary task.

Motivated by the present situation on the type-II Dirac metals, in this work, we investigate the topological properties and the synthesis method of Sn_2_CoS with an L2_1_ crystal structure. We will show that Sn_2_CoS has a type-II Dirac point at 0.15 eV, which is quite close to the Fermi level and available to detect the topological signal in an experiment. The nanowires of Sn_2_CoS are successfully grown by the ECD method. The nanowires are single crystals with an axis direction of [100]. The surface state can play a role in practical devices that are based on the topological electrical transport properties of nanowires.

## 2. Computational Details and Experimental Methods

The band structure calculations are performed based on the density functional theory (DFT), as implemented in the Vienna ab initio simulation package (VASP). The generalized gradient approximation with the realization of the Perdew–Burke–Ernzerhof functional is adopted for the exchange-correlation potential. The cutoff energy is set to 500 eV, and the Brillouin zone is sampled with 11 × 11 × 11 *Γ*-centered *k* mesh. The energy convergence criterion is chosen to be 10^−6^ eV. The surface states are calculated by using the Wannier tools package.

The Sn_2_CoS nanowires were synthesized in an AAO template by direct current electrochemical deposition. The AAO template was used as the working electrode, and Pt as the counter electrode. In the electrochemical deposition synthesis process of compounds, the chemical composition ratio of compounds is determined by the reduction and precipitation rate of metal ion co-deposition, the reduction potential, and the ion mobility of the ion source [48]. The cathode overpotential is the main factor affecting the reduction and precipitation rate. Compared with the standard hydrogen electrode, the reduction potentials of Sn, Co, and S are −0.136 V, −0.277 V, and −0.476 V, respectively. Among them, S has the highest reduction potential, the lowest reduction potential, and the larger ion radius. It quickly migrates to the cathode and reduces the ion mobility of Co ions, resulting in excessive S deposition and Co deficiency. In order to obtain a standard composition ratio of 2:1:1 (Sn:Co:S) with an L2_1_ structure, we adjust the amount of Co source and S source to deviate from the standard ratio during the electrochemical deposition process. The optimized electrolyte contains 0.2 M CoSO_4_·7H_2_O, 0.1 M SnSO_4_, and 0.05 M C_2_H_5_NS for electrochemical deposition. In addition, 0.1 M boric acid (H_3_BO_3_) is added as a buffer to make electrochemical deposition easier, as well as 0.006 M ascorbic acid (C_6_H_8_O_6_) and 0.005 M NaCl, which are added to enhance the conductivity of the solution, respectively. A small amount of sodium dodecyl sulfate (C_12_H_25_NaO_4_S) can act as an active agent. Here, 0.002 M C_12_H_25_NaO_4_S is used as an anionic surfactant to reduce the surface energy and allow better access of the solution to enter the pores of the template. As a complexing agent in the electrolyte solution, sodium gluconate is added to help the electrode potential of ions to gather together.

After preparing the solution required for electrochemical deposition, we performed electrochemical deposition for 1 h by DC constant voltage of −1 V at room temperature. Then, the AAO template with nanowires was taken from the solution to immerse in 2 M NaOH for 6 h. After the dissolution of the AAO template, the achieved nanowires were cleaned in deionized water by ultrasonic cleaning equipment.

We analyzed the structure of the nanowires using X-ray diffraction (XRD) (Cu K_α_) and transmission electron microscope (TEM). A scanning electron microscope (SEM) with energy dispersion spectroscopy (EDS) was used to observe the morphology and detect the chemical composition ratio of nanowires. The high-resolution images (HRTEM) and electron diffraction patterns were achieved by the TEM examinations. The nanowire samples used for SEM and TEM were made by dropping the alcohol suspension of nanowires onto the silicon wafer or copper mesh.

## 3. Results and Discussion

The L2_1_ structure model of Sn_2_CoS is shown in Figure 1a. The positions of the Sn atoms are (1/4, 1/4, 1/4), while the Co and S atoms are located at (0, 0, 0) and (1/2, 1/2, 1/2) in Wyckoff coordinates, respectively. The L2_1_ structure belongs to space group *Fm*$\bar{3}$*m* (No. 225).

Figure 2a shows the band structure of Sn_2_CoS with an L2_1_ structure calculated without spin–orbit coupling (SOC). It obviously shows a metallic band structure with several bands traversing the Fermi level. Very interestingly, as shown in Figure 2a, there are two bands that cross along the *Γ-X-K* path, forming two band-crossing points near the Fermi level (at 0.15 eV), named *P*_1_ and *P*_2_. The crossing points so close to the Fermi level are conducive to the detection of topological signals by angle-resolved photoemission spectroscopy (ARPES) in the future. Noticing that the structure of Sn_2_CoS reserves both the time-reversal (*T*) and spatial inversion (*P*) symmetries, thus *P*_1_ and *P*_2_ are not isolated nodal points but may belong to a nodal line. In order to confirm the characteristics of the nodal line, careful investigation of the band structure near the crossing points *P*_1_ and *P*_2_ was performed by calculating the band structure along the selected four *k* paths, namely *X-a*, *X-b, X-c,* and *X-d* (see Figure 2c). Figure 2b,d show the band structures along the *k*-paths as indicated in Figure 2c and the profile of the nodal loop in Sn_2_CoS. It is clearly illustrated that the crossing points *P*_1_ and *P*_2_ belong to a type-II nodal loop centering the *X* point. For nodal lines, one typical signature is that they can show drumhead-like surface states. To obtain the surface states of Sn_2_CoS, we constructed a first-principles tight-binding model by projecting onto the Wannier orbitals. We calculated the (1 1 0) surface state of Sn_2_CoS, and the results are shown in Figure 2e. We can observe clear drumhead-like surface state originating from the bulk band crossings, as pointed out by the arrows.

Now we turn to the band structure of Sn_2_CoS with SOC considered in the calculation as shown in Figure 3a. With the action of the spin orbit, *P*_1_ and *P*_2_ points are usually into Dirac points. So, in Figure 3a, we rename them as *D*_1_ and *D*_2_ points to distinguish them in discussions. Compared with Figure 2a, it can be observed that the *D*_1_ point still exists, but *D*_2_ is gapped with a small band gap (38.3 meV). Due to the coexistence of *P* and *T* symmetries in the system, the bands are both doubly degenerate. Thus, the band-crossing *D*_1_ forms a Dirac point, as shown in Figure 3a. Figure 3b shows the three-dimensional (3D) band structure about Dirac point (*D*_1_). We can clearly observe a type-II dispersion. Dirac metals usually manifest Fermi-arc surface states. In Figure 3c, we show the surface state band structure on the (1 1 0) of Sn_2_CoS with SOC. The exact positions of the Dirac point (*D*_1_) are indicated and two Fermi arcs (pointed out by the arrows) originating from the Dirac point are observed in the projected spectrum shown in Figure 3c. Through detailed research, we also find that there are seven Dirac points near the Fermi energy level, as shown in Appendix A. Further, we also give a list of more detailed information on all the Dirac points in Appendix A, including the positions in the *k*-space and the energy levels. Based on the above observation and discussion of the theoretical calculation results, it can be predicted that Sn_2_CoS with an L2_1_ structure is a type-II Dirac metal.

Next, we provide more details of the orbital-projected band structure and the density of states (DOS) of Sn_2_CoS in Figure 4. By analyzing the orbital components, we find that two bands forming nodal line mainly originate from the Co-*d* orbital and S-*p* orbital (see Figure 4b). Since PBE computations may underestimate the band gap, we also check the band structure using the modified Becke–Johnson (MBJ) potential. As shown in Figure 5, the results are consistent with those calculated by PBE. In addition, we consider Coulomb interaction (using GGA + U method) and calculate the band structures with U values of 2, 4, 6, 8, and 10, respectively, as shown in Appendix A. In accordance with the actual experimental value, we choose the band structure of Sn_2_CoS without the U value finally, as shown in Appendix A.

Encouraged by the theoretical results, we try to synthesize the Sn_2_CoS with an L2_1_ structure in experiments. Firstly, we use the solution with the standard ratio of 2:1:1 (Sn:Co:S) to prepare the nanowires. It was found that no matter what deposition voltage is used, the Sn_2_CoS with an L2_1_ structure cannot be grown. As described in Section 2, it is impossible to obtain Sn_2_CoS with an L2_1_ structure and the standard composition ratio of 2:1:1 (Sn:Co:S) if the deposition voltage just meets the deposition potential of the three elements but the proportion of various elements in the solution does not match the deposition voltage. Therefore, we adjust the amount of Co source and S source to the chemical composition ratio of 2:4:1 (Sn:Co:S) for the electrochemical deposition process as described in Section 2 of this work.

Proper voltage is necessary for the solution with the corresponding element ratio. Using the solution with the chemical composition ratio of 2:4:1 (Sn:Co:S), the nanowires are grown at different deposition voltages to investigate the relationship between voltage and the composition of nanowires. As typical examples, Figure 6a,c show the XRD patterns of the samples prepared at the deposition voltages of −0.5 V and −1 V, respectively. It can be found that whether at −0.5 V or −1 V, ions can enter holes in the AAO template to form nanowires. However, the difference is that the nanowires deposited at −0.5 V not only deviate from the standard composition ratio of 2:1:1 (see EDX spectroscopy in Figure 6d) but also do not crystallize in L2_1_ structure (see XRD pattern Figure 6c). This is because the precipitation potential of Sn ions has relatively high potential energy when the deposition voltage is −0.5 V, and their deposition rate on the cathode is very rapid, which can easily inhibit the migration rate of Co. So, Sn, and some intermediate compounds (such as SnS) are introduced, which makes the three elements unable to deposit uniformly. As shown in Figure 6c, the diffraction peaks of Sn are observed as the main ones of nanowires on the XRD patterns. When we increase the deposition voltage to −1 V, the principal peaks of the L2_1_ structure, namely, (200), (220), and (400) peaks, can be clearly observed on the XRD pattern of nanowires in Figure 6a. Except for the diffraction peaks of the AAO template substrate (Al_2_O_3_), no secondary phase diffraction peak appears on the XRD patterns. Figure 6b shows the EDX spectroscopy and the chemical composition ratio of nanowires prepared at −1 V. The chemical composition ratio of nanowires is quite close to the standard composition ratio of 2:1:1 (Sn:Co:S). All these results show that the nanowires of Sn_2_CoS with an L2_1_ structure have been successfully grown in the solution with the chemical composition ratio of 2:4:1 (Sn:Co:S) by electrochemical deposition process at −1 V. The lattice constant of nanowires of Sn_2_CoS with an L2_1_ structure obtained from XRD analysis is 6.0 Å, which is consistent with the results achieved by TEM given later. In addition, it should be noted that the nanowires of Sn_2_CoS with an L2_1_ structure show the excellent preferred orientation of [100] along the axis direction, which implies that nanowires are likely to be single crystals.

Figure 6e shows the nanowire array image on the template obtained after etching the surface of the AAO template. It can be clearly seen that these nanowires lean against each other to form bundles. In order to observe a single nanowire, we dispersed the nanowires in an alcohol solution by the ultrasonic method after the template was etched completely. The alcohol solution with the released nanowires was dropped on the silicon wafer, dried, and observed by SEM. Figure 6f shows an image of the nanowires released after dissolving the AAO template. These nanowires are uniform in length and thickness. Their length is about 70 μm. The nanowires were also picked up by copper mesh for TEM observation. With the help of the images of the single nanowire detected by TEM as shown in Figure 7a,b, it can be obtained that the diameter of the single nanowire is 70 nm ± 10 nm, which fits the size of the hole in the AAO template. This indicates that the nanowires are filled with holes in thickness. In addition, it is worth mentioning that the deposition time has an important influence on the length of nanowires in the AAO template. When the deposition time is less than 1 h in our AAO template, nanowires are hidden in holes and cannot be exposed. Additionally, when the deposition time exceeds 1 h, the nanowires will overflow from the holes of the AAO template to gradually form a film-like covering on the surface of the template. The top of the nanowires will be connected together, just like turf, which is hard to separate. All these results indicate that the nanowires grow from bottom to top, not from the hole wall to the hole center. This growth mode is an important guarantee for the growth of nanowires into single crystals. The Sn_2_CoS nanowires are single crystals, which will be confirmed in the following detection of TEM. When the deposition rate of Sn_2_CoS is about 70 μm/h, a deposition time of 1 h is the best nanowire growth time for the AAO template in this work. In addition, it is possible to grow nanowires with different lengths or radii by using AAO templates with different pore sizes or thicknesses within an appropriate deposition time.

Figure 7c shows a high-resolution transmission electron microscopy (HRTEM) observation. The structure of Sn_2_CoS nanowires is captured clearly, and the well-defined lattice fringes with a space of 0.128 nm can be indexed along the [111] crystal direction. According to the space of 0.128 nm, we can figure out the lattice parameter is 6.0 Å, which is consistent with the result from the XRD pattern. The exposed surface is (110) (inset in Figure 7c). We also performed a series of selective area electron diffraction (SAED) patterns along the growth direction of nanowires to confirm that the Sn_2_CoS nanowires are single crystals as shown in Figure 7d,e. These results show that the Sn_2_CoS nanowires are single crystals along the axial direction of [100].

## 4. Conclusions

We synthesized Sn_2_CoS in experiment and theoretically study the topological properties of this material. By first-principles calculation, we find that Sn_2_CoS has type-II nodal-line and drumhead-like surface states near the Fermi level without SOC. When SOC is considered, the nodal line opens the band gap, leaving type-II Dirac point and Fermi-arc surface states. Most importantly, we also synthesized Sn_2_CoS in experiment. In this work, Sn_2_CoS nanowires with the L2_1_ structure are synthesized successfully by the ECD method. The single-crystal nanowires of Sn_2_CoS with a length of about 70 μm, a diameter of about 70 nm, and an axis direction of [100] were achieved. The nanowires grow from bottom to top in the holes of AAO templates. Nanowires with different lengths or radii can be obtained by using AAO with different pore sizes or thicknesses within an appropriate deposition time. Our experimental synthesis and theoretical study of Sn_2_CoS will provide candidate materials for further measurements by angle-resolved photoemission spectroscopy (ARPES) in the future.

## Figures and Tables

**Figure 1 nanomaterials-13-01389-f001:**
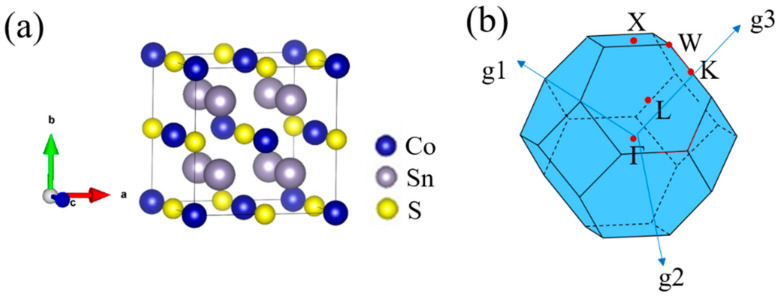
(**a**) Crystal structure and (**b**) the bulk Brillouin zone of the Sn_2_CoS with L2_1_ structure.

**Figure 2 nanomaterials-13-01389-f002:**
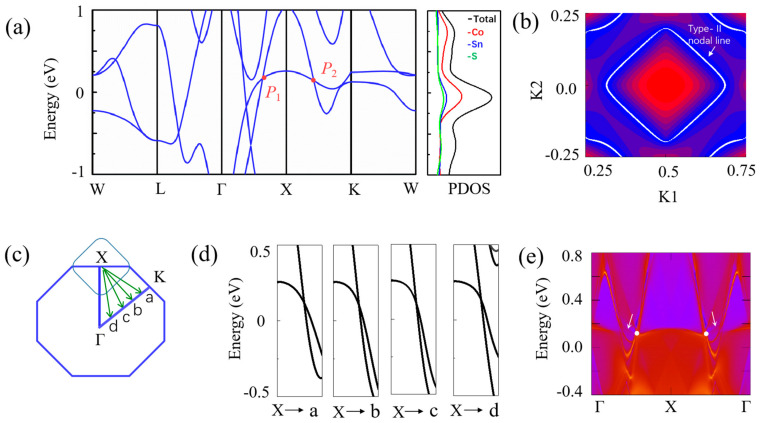
(**a**) Electronic band structure and the projected density of states (PDOS) of Sn_2_CoS without SOC. Two crossing points near the Fermi level along the *Γ*−*X*−*K* path have been labeled as *P*_1_ and *P*_2_, respectively. (**b**) The nodal loop in Sn_2_CoS. (**c**) Illustration of the nodal loop. The points *a*, *b*, *c*, and *d* are equally spaced between *K* and *Γ*. (**d**) Band structures along the *k*-paths as indicated in (**c**). (**e**) The surface state band structure on the (1 1 0) of Sn_2_CoS without SOC.

**Figure 3 nanomaterials-13-01389-f003:**
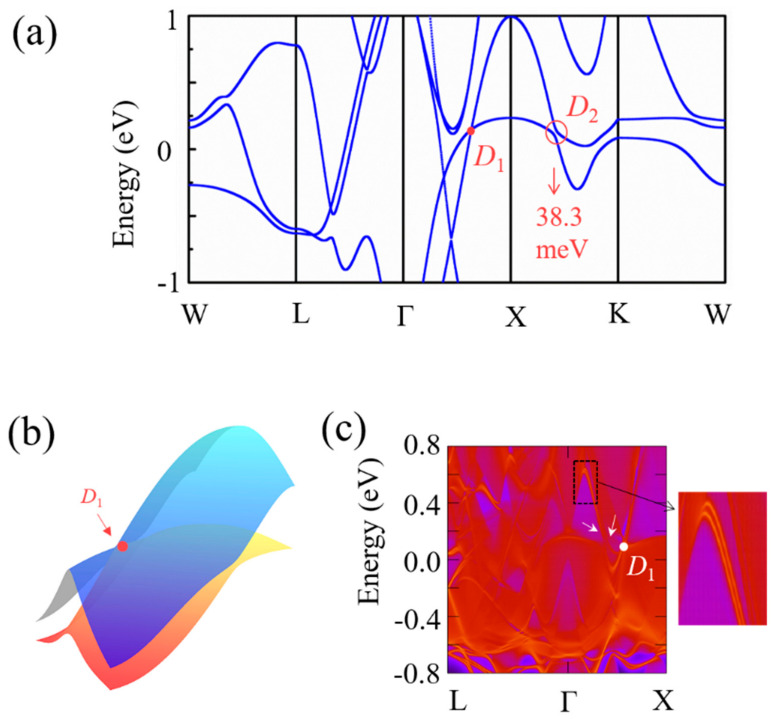
(**a**) Electronic band structure of Sn_2_CoS with SOC. The nodal point and gapped nodal point are indicated by *D*_1_ and *D*_2_. (**b**) The 3D plot of band dispersions near the *D*_1_. (**c**) The surface state band structure of Sn_2_CoS with SOC. The Fermi arcs are pointed by the white arrows.

**Figure 4 nanomaterials-13-01389-f004:**
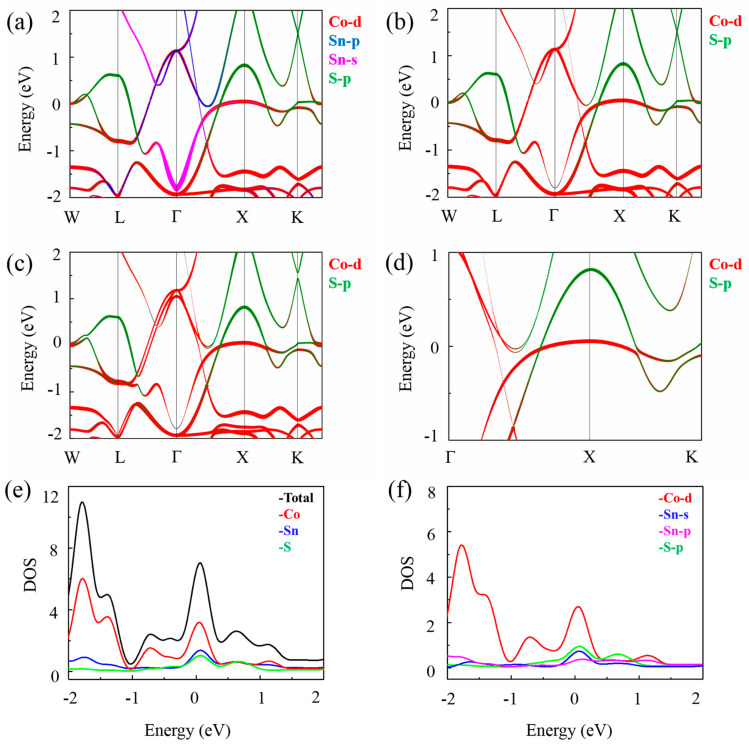
(**a**,**b**) The orbital-projected band structure of Sn_2_CoS without SOC. (**c**,**d**) The orbital-projected band structure of Sn_2_CoS with SOC. (**e**) The density of states (DOS) of Sn atom, Co atom, and S atom. (**f**) The DOS of Co-*d*, Sn-*s*, Sn-*p,* and S-*p*.

**Figure 5 nanomaterials-13-01389-f005:**
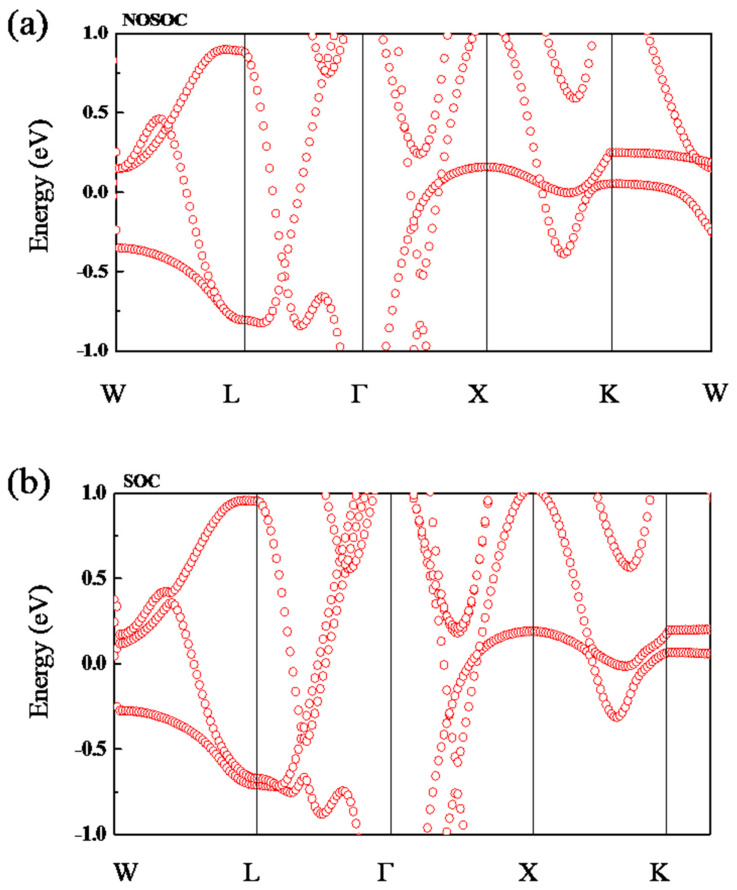
(**a**) The band structure of Sn_2_CoS without SOC by the modified Becke−Johnson (MBJ) potential. (**b**) The band structure of Sn_2_CoS with SOC by the modified Becke−Johnson (MBJ) potential.

**Figure 6 nanomaterials-13-01389-f006:**
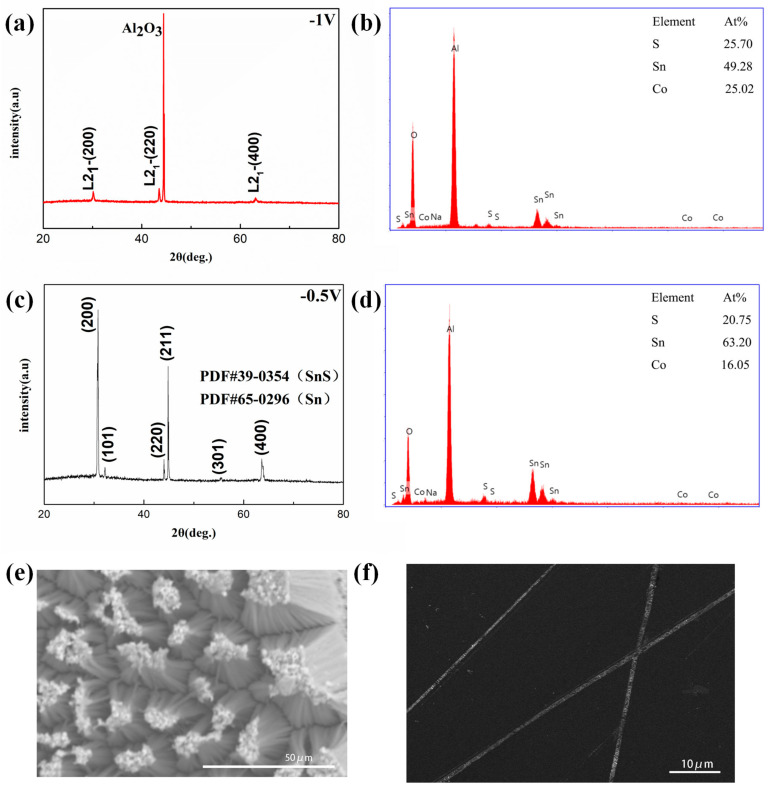
(**a**) The X-ray diffraction patterns of Sn_2_CoS nanowires prepared at −1 V. (**b**) EDX spectroscopy of nanowires at −1 V. (**c**) The X−ray diffraction patterns of Sn_2_CoS nanowires prepared at −0.5 V. (**d**) shows EDX spectroscopy of nanowires at −0.5 V. (**e**) Typical image of nanowire array by SEM. (**f**) The image of single nanowire by SEM.

**Figure 7 nanomaterials-13-01389-f007:**
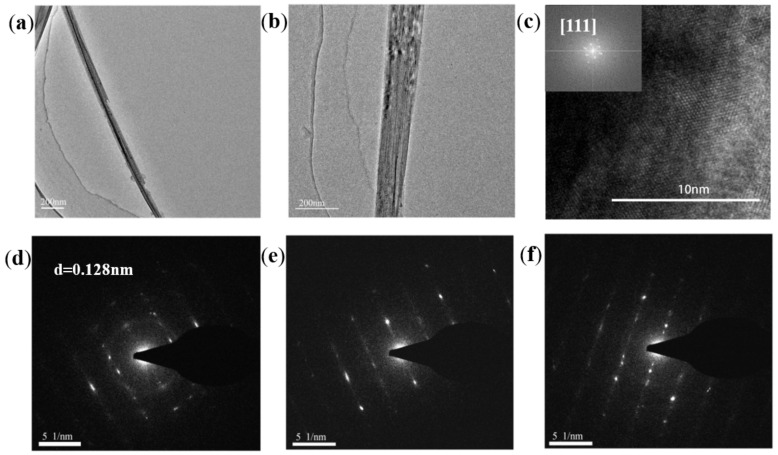
(**a**,**b**) The results of Sn_2_CoS alloy nanowires without AAO template at 200 nm scale by TEM. (**c**) The image of the Sn_2_CoS by HRTEM. (**d**–**f**) The SAED pattern of the Sn_2_CoS single-crystal structure recorded along the [111] crystal orientation.

## Data Availability

Data are contained within the article.

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
