# Peer review of "Nontrivial Topological Properties and Synthesis of Sn2CoS with L21 Structure"

_nanomaterials, 2023, doi:10.3390/nano13081389_

Round 1

Reviewer 1 Report

In this paper, the band structure and surface states of Sn2CoS alloys is investigated by first-principles calculations. A type-II nodal-line near-Fermi level and a drumhead-like surface state indicate that Sn2CoS with L21 structure has nontrivial topological electronic properties, which are also studied in the work. In the experimental section, Sn2CoS nanowires with the L21 structure are synthesized successfully by the ECD method. These materials, featuring the Dirac point which is quite close to the Fermi level, can be further elaborated for topological metals, with promising applications. I find the results interesting and think that the work deserves publication. As a suggestion, the descriptions of the theoretical and experimental parts in the abstract should be linked together better, and the connection between them should be outlined more than just saying that “the NWs have been grown”.      

Reviewer 2 Report

The manuscript presents an ab-initio study and synthesis of Sn2CoS. Using comprehensive DFT calculations, the authors show that the Sn2CoS crystal with L21 structure is a type-II Dirac semimetal. They also demonstrate that Sn2CoS single crystal nanowires can be synthesized by a electrochemical deposition method. The paper is valuable and deserves to be published in Nanomaterials.

Reviewer 3 Report

G. Chen et al. have studies band calculations and experimental synthesis of Sn2CoS with L21 structure. In the band calculation, they find non-trivial topological properties that suggest a type-II nodal topological semimetal. In the experimental part, they successfully grow these Sn2CoS in nanowire forms. 

Unfortunately, the connection between computational work (band calculation) and experimental work is lacking. In the experimental part, there is no evidence supporting what they predict from band calculations. These two works should be separated into two small works, or some additional experimental probe that gives a hint of the band calculations must be provided to justify these two works belonging together. In either case, I cannot recommend this work to be published in a such high-impact journal. 

Reviewer 4 Report

The manuscript presents  both the theoretical calculations and the experimental verification by growth of the proposed material.  It is important, because usually the reliability (or applicability) of first principle calculations is not known in advance. In the present manuscript, authors have obtained the proposed material, which should attract attention of the community to the proposed topological semimetal. 

I have two minor remarks, which could be valuable for the  text improvement.

1. Authors confirm by XRD and electron microscopy the structure of the material. Are any experimental signs of the topological properties? At least, some words should be placed into the text.

2. Fig. 6 (a) shows also huge Al2O3 peak. It is OK, but why there is no Al2O3 in Fig 6 (c)? In the same time, Al is present in both cases in (b) and (d). 

Round 2

Reviewer 3 Report

The authors have not made any improvements in their manuscript. In the author's response, I only see their response to my comments on justifying why this work should be published in this journal.